# Real-Time Interaction for 3D Pixel Human in Virtual Environment

Haoke Deng [1], Qimeng Zhang [2], Hongyu Jin [3] and Chang-Hun Kim [1,*]

1  Department of Computer Science and Engineering, Korea University, Seoul 02841, Republic of Korea
2  BK21 FOUR R&E Center for Computer Science and Engineering, Korea University,
   Seoul 02841, Republic of Korea
3  Interdisciplinary Program in Visual Information Processing, Korea University, Seoul 02841, Republic of Korea
*  Correspondence: chkim@korea.ac.kr

**Abstract:** Conducting realistic interactions while communicating efficiently in online conferences is highly desired but challenging. In this work, we propose a novel pixel-style virtual avatar for interacting with virtual objects in virtual conferences that can be generated in real-time. It consists of a 2D segmented head video stream for real-time facial expressions and a 3D point cloud body for realistic interactions, both of which are generated from RGB video input of a monocular webcam. We obtain a human-only video stream with a human matting method and generate the 3D avatar's arms with a 3D pose estimation method, which improves the stereoscopic realism and sense of interaction of conference participants while interacting with virtual objects. Our approach fills the gap between 2D video conferences and 3D virtual avatars and combines the advantages of both. We evaluated our pixel-style avatar by conducting a user study; the result proved that the efficiency of our method is superior to other various existing avatar types.

**Keywords:** virtual avatar; virtual meeting; pixel-style art





## 1. Introduction

Based on the growing popularity of online meetings, meeting forms have developed rapidly. The two main forms of virtual meetings are 2D and 3D meetings. A 2D video conference platform, such as Zoom [1], which is widely used for online meetings, adopts a pure 2D video stream that can realize real-time video conferencing for multiple people through video streaming and voice chat. However, such 2D video conferences lack the realism of conducting a meeting in a 3D space and cannot meet the needs of participants in terms of displaying 3D virtual objects or interacting with virtual objects in 3D. Therefore, to improve upon traditional 2D video meeting technology, 3D virtual meeting platforms have developed rapidly.

In 3D video conference platforms, such as Spatial [2], users can participate in a conference in the form of a 3D virtual avatar in a virtual meeting room on multiple platforms, such as desktop and mobile devices. In Spatial, users use a single face photo input to form their own avatar's 3D head model and automatically perform general facial animations, such as blinking. However, the facial movements of the avatar are not directly related to the real-time facial actions of the user, and it is difficult to convey expressions and emotions. Spatial also realizes the capture of user arm and hand movements in virtual meetings and allows users to interact with virtual objects. However, these functions are based on virtual reality (VR) glasses and tracking sensors, which require less common equipment and application scenarios. Additionally, the body, skin tone, and clothing of the avatar are generated based on direct selections by users, which makes it difficult to reflect the characteristics of real clothing and body shapes.

The 3D virtual avatar used in the immersive VR system can accurately obtain the motion and position of the user's hands and arms through the VR controller or sensor so as to realize the interaction with the virtual object. However, while using a VR headset

for an online conference, due to the occlusion of the face by the headset itself, it is difficult to represent the user's facial expression in a real-time conference, thereby reducing the effectiveness of communication. In this work, we use a normal monocular RGB webcam, the same as those used in traditional 2D video conferences to include both real-time facial expressions and motions of users in the generated pixel avatar. Meanwhile, unlike the 2D video conference, the pixel avatar also has 3D arms that can be used for virtual interaction, which solves the difficulty of displaying and interacting with 3D virtual objects in online conferences.

Inspired by VirtualCube [3], which visualizes a depth map captured by multiple Azure Kinect [4] devices to form a 3D human with the user's 3D geometry and texture, we propose a pixel-style virtual avatar created from a single monocular RGB webcam to represent a user's body as a 3D point cloud with texture. We follow methods, such as those used in Minecraft [5] and PixelArt [6], to generate a low-resolution pixel-style image from a high-resolution photo while retaining some features of the input images, such as the clothing. Additionally, we provide pixel-style avatars at various resolutions for selection while ensuring real-time performance.

To ensure that a user avatar generated in a 3D video conference does not contain background components that are irrelevant to humans, we adopted the human matting method to obtain segmented human results that only include a user's 2D video stream. To achieve real-time 3D avatar arm movements and virtual object interactions similar to those in Spatial [2], we employed a 3D pose estimation method to obtain the 3D pose information and distribute a point cloud based on a 3D skeleton to convert 2D avatar arm movements into a 3D space. Additionally, real-time arm motion estimation can enhance user interactions.

In a video conference, facial expressions are also essential for efficient communication among users. If the entire video stream of a user is converted directly into a relatively low-resolution pixel-style avatar, subtle expressions and facial movements may be lost. Therefore, in the proposed framework, we only transform the user's body into a pixel style and retain a high-resolution facial video stream to facilitate efficient real-time communication in online meetings.

The main contributions of this study can be summarized as follows:

- We propose a novel pixel-style avatar to solve segmented human real-time 3D visualization problems in a virtual environment.
- We use 3D pose data to generate a 3D pixel avatar arm and use high-resolution facial video to reflect facial expressions, enabling more realistic interactions and intuitive communication.

## 2. Related Works

### 2.1. Avatars in Virtual Meetings

Many existing 3D video conference systems that pursue realism use RGBD cameras to capture a user's 3D geometry and texture. For example, VirtualCube [3] adopts a method that uses multiple Azure Kinect RGBD cameras [4] to capture humans and display remote participant videos at full scale on monitors to improve meeting realism. Other depth sensors, such as Intel RealSense [7], can also visualize humans in 3D using depth maps. However, it is difficult for users to use multiple sensors for video conferencing because high-precision depth sensors have not been widely adopted. Therefore, we aimed to achieve similar depth sensor results for generating 3D point cloud avatars using a general monocular webcam. Spatial [2] utilizes a Metaverse virtual space, where users can create 3D virtual avatars containing their faces and collaborate anywhere using VR, desktop, and mobile devices. The face of the avatar in the virtual space is based on 3D reconstruction from a single photo and facial expressions are simple preset animations. The hand movements of VR users are captured by the sensors in Oculus Quest [8] VR glasses, and those of users on other platforms are simply preset animations. The body skin tone and clothing of avatars in the 3D space do not reflect the actual conditions of users.

AudienceMR [9] provides a multi-user mixed-reality space that extends the local user space into a large, shared classroom for lecturers. A user's full-body avatar in AudienceMR is a 3D avatar whose face is generated from an image of the user. Analysis in AudienceMR has revealed that 3D avatars provide a more realistic feeling in an actual lecture compared to a 2D video stream, similar to our study on virtual conferences. However, user avatars in AudienceMR cannot reflect their clothes, expressions, movements, etc., and are not generated in real-time.

Pakanen et al. [10] summarized 36 comparable human-like virtual avatars used in augmented reality, VR, and virtual spaces, and proposed that the photorealism of an avatar increases the feeling of co-presence. Although we recognize that full-body avatars are the most popular choice for representing users, because we aim to improve traditional 2D video conferencing, where most user cameras only show the upper body, our avatars only include the head, arms, and torso, according to the field of view of a typical webcam. Our work does not use sensors, VR headsets or other special equipment, such as VirtualCube [3] and Spatial [2], except for a common webcam, but is able to provide a similar interactive experience.

### 2.2. Pixel-Style Art

PixelArt [6] uses the CycleGAN [11] style transfer model to convert images or photos into a pixel art style, but it is difficult to transform video streams into a pixel style in real time. PixelMe [12] is a pi2pix-based [13] machine learning model for style transformation that generates 8-bit-style portraits. In this study, we focused on pixelating images containing bodies, not only portraits. The Minecraft Image Converter [5] can convert images into Minecraft blocks, but is difficult to apply to other platforms. Iseringhausen et al. [14] introduced a style transfer method that uses wood samples and a target image as inputs, and generates a cutting pattern for parquetry puzzles. The result is a style-transferred image with a wood texture, but this process is complex and time-consuming.

In this work, we adopt pixel-style 3D point clouds with user textures to represent human bodies while generating avatars. We use an algorithm-based method to process pixel-style images and use particles to represent the pixels of them, making pixel avatars that look more natural in virtual scenes.

### 2.3. Human Matting and Segmentation

Similar to the function of replacing the background in a 2D video stream, we must also remove the background from user video stream inputs when forming a virtual avatar to obtain a clean video stream containing only a human. Commonly used methods for extracting humans from real-time video streams during video conferencing include video matting and semantic segmentation. The methods in [15–17] are based on semantic segmentation methods. In addition to humans, they can also segment other subjects. However, the segmentation results typically contain hard edges and artifacts; therefore, there is a clear boundary between the human image and the background, which is inappropriate for a smooth transition with the background image or virtual environment. In contrast, the results of matting methods had soft boundaries. Methods such as BGM [18] and BGMv2 [19] use a fixed monocular webcam to accomplish real-time high-precision human matting. However, background-based matting methods require an additional pre-captured background image as an advanced input. Trimap-based matting methods, such as those proposed by Xu et al. [20] and Forte et al. [21], rely on additional input trimaps for foreground and background prediction. Zhou et al. [22] proposed a method for automatically generating a trimap for matting, but the trimap generation process is relatively complex. MODNet [23] is an auxiliary-free matting method that operates on frames as independent images, but has difficulty handling fast-moving body parts. Lin et al. [24] and Bodypix [25] proposed multi-person part segmentation methods that can segment human body parts and determine the range of the arms, but they are not sufficiently robust and are relatively time-consuming.

Robust video matting (RVM) [26] is a robust, real-time, high-resolution human video matting method that exploits the temporal information in videos. RVM uses the backbone proposed in MobileNetV3 [27] and uses a deep guided filter [28] to upsample video resolution. Because we intend to mat humans from backgrounds in real-time when forming a virtual avatar and hope that our application can be used directly without additional preparation or inputs, in this study, we adopt the RVM method to segment humans from backgrounds.

*2.4. Pose Estimation*

Pose estimation is a method for predicting and tracking the location of a person or group of people and obtaining 2D or 3D landmarks. Recently proposed 3D monocular pose-estimation methods, such as those in [29,30], can perform three-dimensional human pose estimation with high accuracy and robustness, but because these methods use the Human3.6M dataset proposed by Ionescu et al. [31], they all estimate a relatively small number of joints. In particular, the results predicted by these two methods do not contain certain joints in the human face and hands that are required by our method to generate 3D avatars. Therefore, they are not applicable. Additionally, although we only use a few joints in the face, arms, and hands, in the future, when expanding our method to whole-body 3D avatars, a higher number of human joints will yield better results and be more scalable.

MediaPipe Pose [32,33] uses a blazePose [34] detector and the pose landmark model proposed by Xu et al. [35]. It is a relatively accurate real-time 3D pose estimation library that can obtain the 2D and 3D coordinates of 33 key points. In this study, we used 3D pose coordinates from MediaPipe Pose to convert 2D pixel human arms into 3D. Additionally, the MediaPipe Pose only recognizes the pose data of the object with the highest probability of being a human, which helps us convert only the users participating in a video conference into three dimensions while ignoring other objects.

## 3. Pixel-Style Avatar Generation Method

Our method consists of two main components. The first component is video stream processing based on input monocular RGB video for obtaining human segmentation results by using the matting method and parsing the human body region into a high-resolution head and low-resolution body video stream. Simultaneously, pose estimation is adopted to extract 3D pose data. The second component of visualization uses the results of video processing to display a 2D high-resolution head video, generate a point cloud based on low-resolution body video data, and use 3D pose data to convert the arm regions into three dimensions. Finally, based on the pose data, the video stream and point cloud regions are combined to form a complete pixel avatar. Figure 1 illustrates the architecture of the proposed framework.

We describe the basic matting and pose estimation methods used in our framework in Section 3.1. Section 3.2 describes the details of human video stream processing, and Section 3.3 introduces the visualization method used in our approach for generating final pixel-style avatars.

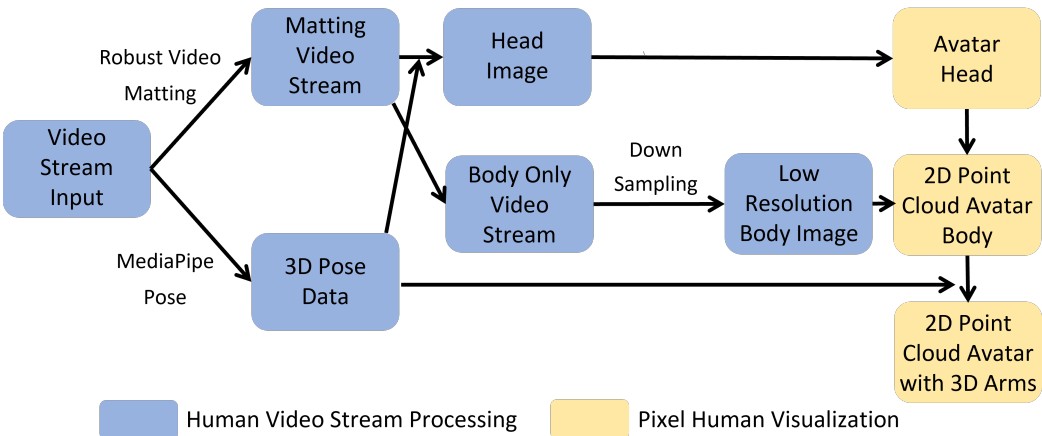

**Figure 1.** Our framework consists of two components, namely video stream processing and visualization. The video stream processing component generates a high-resolution segmented head and low-resolution body video, and extracts 3D pose data. The visualization component uses the results of the video stream processing to visualize a pixel-style avatar.

### 3.1. Matting and Pose Estimation Methods

When generating a pixel avatar for virtual environments, we require a human-only video stream and a smooth transition between the input video and virtual environment. To achieve this goal, we adopt the real-time auxiliary-free matting method RVM [26] to segment users from backgrounds in input video streams to obtain foreground images containing only humans and alpha mattes. To obtain 3D pose data for partitioning human body parts and visualizing an avatar, we use MediaPipe Pose [29,33] for real-time pose estimation. MediaPipe Pose is a machine learning solution for high-fidelity body pose tracking, predicting 2D frame relative coordinates normalized to [0.0, 1.0], and representing joint positions on a screen and 3D real-world coordinates in units of meters for 33 human body joints. Figure 2 presents an original input frame and the corresponding results of the RVM and MediaPipe Pose methods.

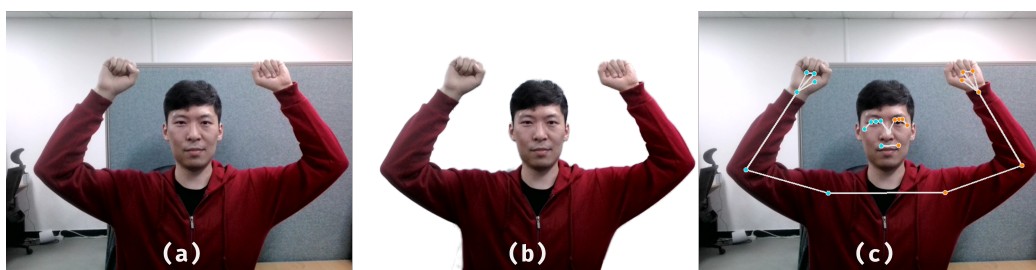

**Figure 2.** Results of RVM and MediaPipe Pose. (**a**) Original input frame of the upper body in 640 × 480 resolution. (**b**) RVM output without background. (**c**) Upper-body joints from MediaPipe Pose.

### 3.2. Human Video Stream Processing

We use the type of monocular RGB webcam that is generally used in online meetings as an input device and applied the RVM to each frame to obtain human matting results. According to the RVM [26], a frame $I$ can be considered a linear combination of the foreground $F$ and background $B$ based on a transparency coefficient $\alpha$ as follows:

$$I = \alpha F + (1 - \alpha)B \tag{1}$$

Because only the human region is required for our virtual environment, we only retain $\alpha F$ for the foreground while omitting the background part $(1 - \alpha)B$ to obtain a final human-only segmented video frame $I$ in four RGBA channels. Based on frame $I$, we parse

the human video frame into high-and low-resolution regions. Because high-resolution video can express facial details to enhance the communication experience during a video conference, we define the head region of the user as having high resolution. Because body features such as cloth color with large regions can be represented roughly, for efficient computation, we define the body region of the user as having low resolution.

To partition the head region in a human video efficiently, we generate an adaptive head range based on extracted 3D pose landmarks. Figure 3 presents the head region and relative pose landmarks. We specify that the range of the head is a square with the nose (blue point in Figure 3) as the center point and twice the 2D relative distance from the neck joint to the nose joint in the frame as the side length. Because the neck joint is not included in the 33 joints of MediaPipe Pose, we calculate the midpoint between the left and right shoulder joint 2D coordinates as the position of the neck joint. We calculate the square side length of the head based on the position of the nose and neck, and then crop the head region from the high-resolution human-only segmentation results. The final head region corresponds to the red ox in Figure 3.

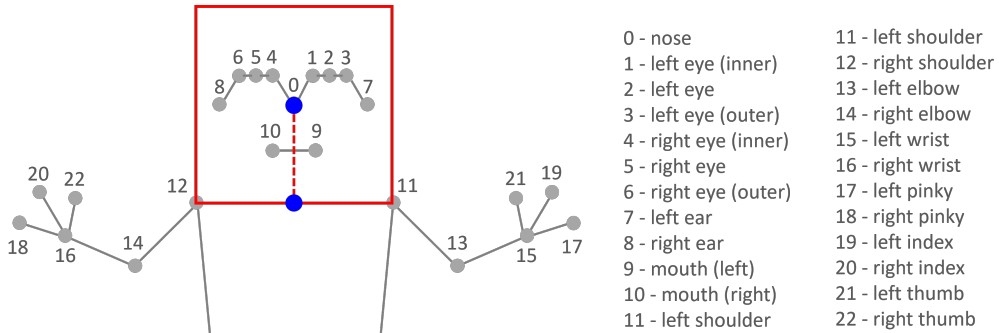

| | |
|---|---|
| 0 - nose | 11 - left shoulder |
| 1 - left eye (inner) | 12 - right shoulder |
| 2 - left eye | 13 - left elbow |
| 3 - left eye (outer) | 14 - right elbow |
| 4 - right eye (inner) | 15 - left wrist |
| 5 - right eye | 16 - right wrist |
| 6 - right eye (outer) | 17 - left pinky |
| 7 - left ear | 18 - right pinky |
| 8 - right ear | 19 - left index |
| 9 - mouth (left) | 20 - right index |
| 10 - mouth (right) | 21 - left thumb |
| 11 - left shoulder | 22 - right thumb |

**Figure 3.** The head image is a square image with the nose as the center and twice the length from the nose to the neck as the side length.

For the low-resolution body region, to reduce the computational complexity and realize real-time performance, the body parts in the high-resolution human body-only video stream are downsampled by a factor of $\frac{1}{2}$ $\frac{1}{4}$ $\frac{1}{8}$ or $\frac{1}{16}$ of the original video resolution by applying the Gaussian pyramid approach. The results are then represented in the form of a point cloud in the visualization process.

### 3.3. Pixel Human Visualization

As mentioned previously, our pixel avatar consists of a head video frame and point cloud body. Additionally, to achieve a realistic interaction experience, we convert the point cloud of the arm region into 3D to obtain a final interactive 3D pixel avatar.

### 3.3.1. 2D Head Visualization

Head visualization directly uses the high-resolution head images obtained in Section 3.2. To demonstrate the necessity of using a high-resolution head image, we generate and compare a pixel avatar with a high-resolution head and a pixel avatar with a low-resolution head matching the resolution of the body. As shown in Figure 4a, when both the head and body of the avatar are downsampled to the same low resolution to form a point cloud, the styles of the two parts are more consistent. However, based on low resolution of the head, it is difficult for the pixel avatar to reflect the user's real-time facial expressions and actions, which may affect communication efficiency during online video conferences. In contrast, the proposed avatar with a high-resolution head in Figure 4b has the advantage of enhancing the communication experience, similar to a 2D video conference. Therefore, we adopt a high-resolution head video stream for the head of a pixel avatar.

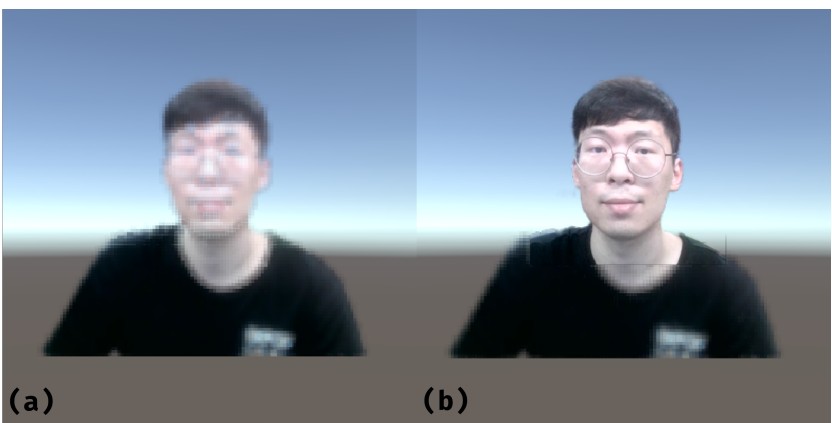

**Figure 4.** (**a**) Low-resolution point cloud head. (**b**) High-resolution 2D plane head.

The obtained high-resolution head images are displayed in a virtual space in the form of a 2D plane that forms the head of a pixel avatar. To control the avatar head movement, we consider the 3D coordinates of the nose joint in the virtual space as the center point of the plane and move the head plane according to the nose joint position in the video stream. To ensure that the ratio of the 2D head plane matches the ratio of the point cloud body in the pixel avatar, we scale the head plane according to the ratio in the original frame.

### 3.3.2. Point Cloud Body Visualization

Body part visualization considers each pixel in the low-resolution body images obtained by downsampling in Section 3.2 and represents it as a particle of in a point cloud. We realize that when directly visualizing the low-resolution human-body-only segmented result as a point cloud, the edge of the point cloud appears unnaturally gray, as shown in Figure 5a. This is because we only retain the coefficient $\alpha$ of the human foreground $F$ in Equation (1) and discard the coefficient $(1 - \alpha)$ of the background $B$ when combining components. Because the background of the virtual environment changes over time, it is difficult to use the real-time background image in the virtual space and combine it with human-only segmented results according to Equation (1) using the coefficient $(1 - \alpha)$. Therefore, we directly add the virtual space background $B_v$ to the human-only segment result $\alpha F$, and the combined frame $I_v$ becomes

$$I_v = \alpha F + B_v \tag{2}$$

A greater combination coefficient for the background $B_v$ results in a larger combined alpha value along the edges, making the edge color appear darker. Gray edges are particularly noticeable when visualizing low-resolution images. Therefore, to eliminate the gray edges around the avatar after converting it into a point cloud, we propose a method to weaken and remove such edges.

First, we process the low-resolution segment result using Laplacian edge detection and subtract the edge detection result from the alpha channel. Figure 5b presents the results after subtracting the detected edges, where most of the gray edges are removed. Subsequently, the histogram equalization method is applied to the alpha channel to soften the sense of the edge further. Figure 5c reveals that the pixel avatar blends smoothly with the virtual space after alpha-channel histogram equalization. Finally, the result of the edge-processing method is converted into a point cloud, and we can obtain an avatar body that does not have strong edges separating it from the surrounding virtual environment.

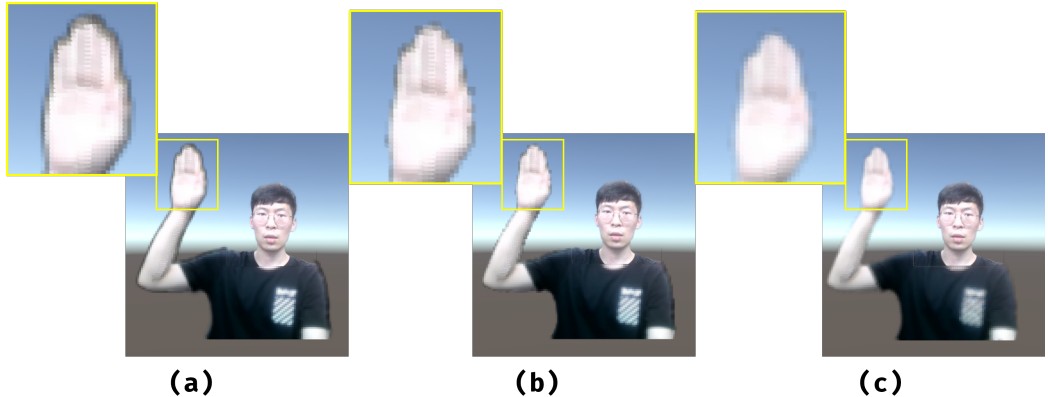

**Figure 5.** (**a**) Downsampling-only body point cloud. (**b**) Result of subtracting the edge detection results from the alpha channel of (**a**). (**c**) Histogram equalization on the alpha channel of (**b**).

### 3.3.3. Interactive Arm Generation

Our study aimed to improve traditional 2D video conferencing applications, where a fixed monocular RGB webcam typically only shows the upper body. Therefore, in this study, we only convert the arms into 3D from an upper-body video stream and realize the function of avatar arm movement according to the actual movements of users.

While generating 3D interactive arms, depth maps by real-time depth estimation methods such as [36,37] are not accurate enough to represent the body details, which affects the results while converting the arms into 3D. The high-precision depth map predicted by [38–40] can more accurately represents the 3D position and shape of human arms. It can also assist pose estimation in obtaining a more accurate arm range. However, these high-performance depth estimations are highly time-consuming and often used for processing pictures or non-real-time videos rather than real-time applications, such as ours. Human matting and pose estimation are necessary for our real-time pixel avatar generation method. Additional depth estimation method will lead to a sharp drop in the speed of generating avatars, affecting the interactive experience. Therefore, in this work, we use the results of 3D pose estimation to convert the arms of the pixel avatar into 3D according to the 3D skeleton rather than using depth estimation.

When determining the arm and hand range that must be converted to 3D, we use the starting and ending points of the upper arms, lower arms, and hands in the virtual space. As shown in Figure 6, the starting point of the upper arm is the shoulder joint, the ending point is the elbow joint. The starting point of the lower arm is the elbow joint, and the ending point is the wrist joint. The starting point of the hand is the wrist joint and extends along the index joint.

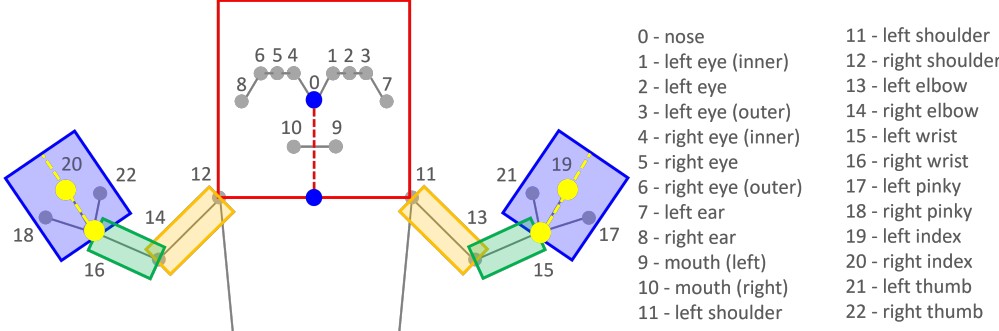

| | |
|---|---|
| 0 - nose | 11 - left shoulder |
| 1 - left eye (inner) | 12 - right shoulder |
| 2 - left eye | 13 - left elbow |
| 3 - left eye (outer) | 14 - right elbow |
| 4 - right eye (inner) | 15 - left wrist |
| 5 - right eye | 16 - right wrist |
| 6 - right eye (outer) | 17 - left pinky |
| 7 - left ear | 18 - right pinky |
| 8 - right ear | 19 - left index |
| 9 - mouth (left) | 20 - right index |
| 10 - mouth (right) | 21 - left thumb |
| 11 - left shoulder | 22 - right thumb |

**Figure 6.** The upper arm range is represented by the orange box from the shoulder joint to the elbow joint and extending to both sides. The lower arm range is represented by the green box from the elbow joint to the index joint and extending to both sides. The hand range is represented by the blue box extending from the wrist joint along the index joint to the first pixel with an alpha value of zero.

Because the coordinates of the 3D pose landmarks obtained by pose estimation have different coordinate systems and units compared to those of the avatar joints in the virtual space, we must obtain suitable coordinate transformation parameters. When generating a 2D point cloud avatar, we already obtain the x- and y-axis coordinates of the joints in virtual space, so we only need to know the transformation parameters of the z-axis coordinates. Because the 3D world coordinates of MediaPipe are all in meters, we can calculate the scaling parameters $z_{scale}$ between the z-axis coordinates in virtual space and z-axis coordinates in world coordinates according to the scaling parameter of the x- and y-axis coordinates. The equation for $z_{scale}$ is as follows:

$$z_{scale} = \frac{\left| x_{left\_shoulder} - x_{right\_shoulder} \right|}{\left| X_{left\_shoulder} - X_{right\_shoulder} \right|} \tag{3}$$

where $x_{left\_shoulder}$ and $x_{right\_shoulder}$ are the coordinates of the left and right shoulders in the virtual space, and $X_{left\_shoulder}$ and $X_{right\_shoulder}$ are the world coordinates of the left and right shoulders, respectively. By scaling the z-axis coordinates of the real-world pose landmarks according to $z_{scale}$, we can obtain virtual space world coordinates of the same scale in all directions.

For each part of the arm, we start from the 3D virtual space position of the starting point to perform 3D steps in the direction of the end position and form a one-pixel-wide skeleton from the starting point to the ending point. We then expand from the skeleton pixels to both sides and set the z-axis coordinates for the pixels based on the scaling parameter $z_{scale}$ to realize a 3D arm.

We define an adaptive parameter $P_w$ for the expanded width of the arms and hands of users with different body sizes and distances from the webcam. First, we calculate a positively correlated arm width $Width_{min}$ based on the 2D distance between the two shoulders in the video frame and set $Width_{min}$ as a minimum threshold to prevent the 2D relative distance between the two shoulders from approaching zero when the person is sideways relative to the webcam. For the adaptive arm width $Width_{arm}$, the width begins to increase from the threshold of the minimum arm width $Width_{min}$ obtained previously and determines whether the alpha values of the pixel at the arm width position on both sides of the center pixel are zero. The alpha values of the pixels on the left and right sides of the skeleton are identified as $\alpha_{left}$ and $\alpha_{right}$, respectively. If either of them is not zero, then we continue to increase the arm width $Width_{arm}$ based on the parameter $P_w$, which has a value of one pixel, until both the values of $\alpha_{left}$ and $\alpha_{right}$ are zero. We specify that the arm width $Width_{arm}$ that achieves this condition is the width of the arm corresponding to the originating pixel on the skeleton:

$$Width_{arm} = Width_{min} + P_w, \ Width_{arm} < Width_{max}$$
$$P_w = P_w + 1, \ \ if \ \alpha_{left} > 0 \ or \ \alpha_{right} > 0 \tag{4}$$

Simultaneously, we set an upper limit $Width_{max}$ for the width of the arm to prevent the arm width from becoming too large, such as when the arm is in front of the body. By using this adaptive arm width method, we can obtain different widths for each position along the arm in real time according to the shape of the arm so that the thickness of each part of the pixel avatar arm varies to mimic the shape of a real person.

Because the 33 joints in a MediaPipe pose do not include the finger joints and only contain three joints on the palm, it is difficult to use the same method of start and end points used for the upper and lower arms to determine the range of the hands. For the hands, when generating a one-pixel-wide skeleton, we begin from the wrist joint and extend it in the direction of the index joint until reaching the first pixel with an alpha value of zero. We use the distance from the wrist joint to this pixel as the length of the hand. The hand is then extended adaptively to both sides of the skeleton pixel in the same manner as the arm to obtain the hand area to be converted into three dimensions.

After the widths of arms and hands are determined, the extended particles are retracted to approximate the curvature of an ellipse, which is a circle stretched 1.5 times in the front-to-rear direction, resulting in a three-dimensional effect similar to an arm cylinder. To solve the particle discontinuity problem caused by elliptical curvature, we set the size of the extended particles to increase linearly, meaning the particles on both sides are larger. In Figure 7, the yellow center pixel represents the skeleton pixel.

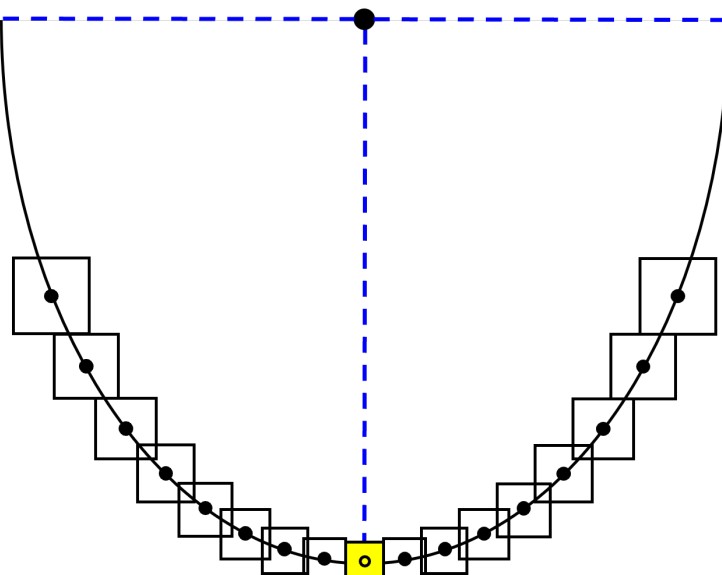

**Figure 7.** Extension from the center skeleton pixel to both sides and retracting to approximate the curvature of an ellipse. The edge lengths of the particles increase linearly from the center to the sides. The yellow pixel is the skeleton pixel.

Additionally, the elbow region requires a natural transition to solve the connection issue when the upper and lower arms fold together. After converting the upper arm into three dimensions, we draw a point cloud in the shape of a quadrangular pyramid extending backward from the 3D position of the elbow joint point. We then convert the lower arm into 3D. We follow the order of the upper arm, elbow, lower arm, and hand when generating the avatar's 3D arm to prevent the loss of parts of the arm.

After applying the basic framework of pixel avatar generation, we conduct an additional process to address the issue of missing 3D arm information. When some joints of the user's arms do not appear in the frame because of the webcam's field of view, the pose estimation of the joints outside the frame sometimes becomes inaccurate. Suppose we directly draw a 3D arm based on incorrect arm joint coordinates outside the screen predicted by pose estimation. The part of the arm outside the screen has no tone and texture, so the point cloud color corresponding to the arm cannot be obtained. Therefore, when we determine whether the avatar's arm needs to be converted into three dimensions, we adopt the principle of skeleton drawing from MediaPipe poses. If the starting and ending points of an arm part are both within the scope of the webcam's screen, then that arm part is converted into three dimensions. As shown in Figure 6, the upper arm's starting point is the shoulder joint, and its ending point is the elbow joint. The lower arm's starting point is the elbow joint and its ending point is the index joint. The avatar's hand uses the same logic for judgment with the wrist joint as the starting point and the index joint as the ending point.

## 4. Implemented Experiences

The main goal of the proposed pixel-style avatar is to enhance communication and interaction experiences during virtual meetings. To achieve this goal, we implemented a virtual meeting room scene containing interactive 3D objects to verify that pixel-style

avatars can improve realism and enhance the virtual interaction experience. Additionally, we implemented two camera perspectives to observe the virtual environment. The first-person view camera of the pixel avatar was placed at the position of the nose joint so that the camera view could change according to the movement of the user's head, similar to a real-world view. Additionally, through the first-person camera, users can see their own 3D pixel avatar arms so that more realistic and interactive virtual object interactions can be realized. A third-person camera was also included to provide a free viewpoint for observing the entire virtual space.

Based on this virtual meeting scene, we conducted two main experiments to investigate the proposed avatar. The first experiment compared the user experience with the proposed avatar to those with two types of existing 2D and 3D avatars. Additionally, we implemented an external skeleton avatar based on our employed human segmentation method and pose estimation for fair comparisons to the baselines. The second experiment was designed to confirm the usability of the proposed pixel-style avatar. The details of our experiments and analyses are described in Section 5.1.

## 5. Experiments and Results

### 5.1. Environments and Participants

Our virtual meeting scene was implemented using Unity 2020.2.7f1 on a PC with a 3.70 GHZ Intel i7-11700K CPU [41] and Nvidia GTX 3080Ti GPU [42]. The webcam used in the experiment is the ABKO [43] APC930 QHD.

For our experiments, we recruited 16 unpaid participants (6 female and 10 male) ranging in age from 23 to 31 years with an average age of 26.75 years. All participants in the experiment had the same experimental items. Experiments were conducted in a virtual meeting room that we constructed, and the equipment required for all methods consisted of an RGB monocular webcam.

### 5.2. Comparative Experiment

In our comparative experiments, we compared our method, which is shown in Figure 8(a1,a2), to three baseline methods. The first baseline method is a traditional 2D video conference, as shown in Figure 8(b1,b2). Because a typical 2D video conference containing background information is too obtrusive in a virtual environment and could affect user experiences, we used the RVM [26] to remove the background and used human-only live video streaming in the 2D video conference. The second baseline method is an artificial avatar, as shown in Figure 8(c1,c2). We used MediaPipe's 3D pose landmark to enable the artificial avatar to follow human movements in real time through inverse kinematics. The third baseline method was a skeleton avatar, as shown in Figure 8(d1,d2). This approach adds a 3D arm skeleton generated from MediaPipe's 3D pose landmarks to the 2D video conference baseline. The joints are represented as spheres and the parts connecting joints are represented as rods. When conducting our experiments, our method used $\frac{1}{4}$ of the original video resolution ($640 \times 480$), resulting in an avatar resolution of $160 \times 120$.

All participants performed the experiments using PC as the user interface, which adapted a monocular webcam as the video stream input. The comparative experiments involved interacting with a virtual object in a virtual meeting room from the first-person and third-person perspectives. The first-person view is how users see their arms in scenarios interacting with virtual objects. The third-person perspective is how a user appears to other participants in a virtual interaction scenario. The interactive virtual objects included a cup and laptop that could be raised by avatar hands, and a toy duck that could be moved and rotated by avatar hands. Our subjects performed virtual object interaction experiments in the first-person and third-person perspectives for all four methods, as shown in Figure 9, and scored the four methods from one to five points based on three questions, with one point representing the worst experience and five points representing the best experience.

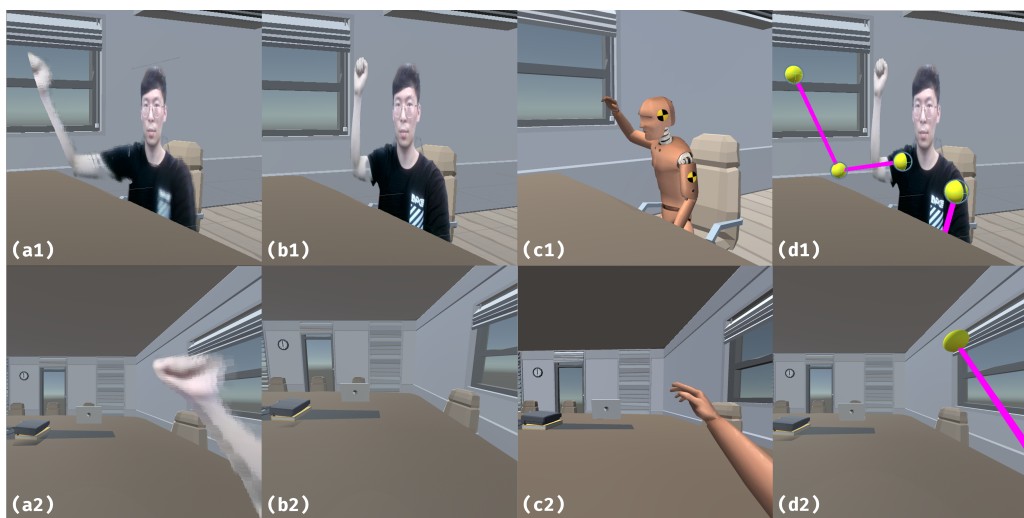

**Figure 8.** Different methods in the comparative experiments. (**a1,a2**) Pixel avatar. (**b1,b2**) 2D video conference. (**c1,c2**) Artificial avatar. (**d1,d2**) Skeleton avatar. The first row is the third-person view, and the second row is the first-person view.

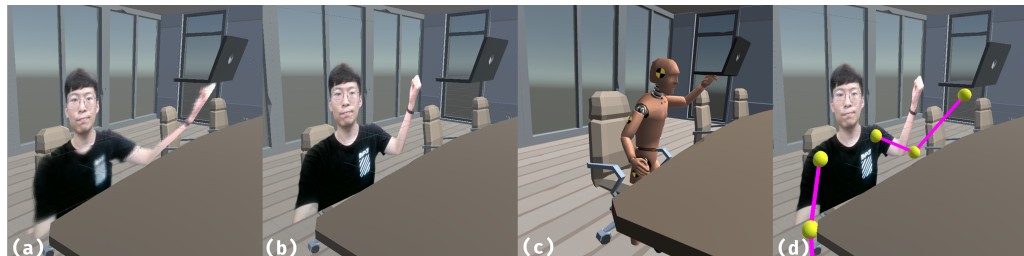

**Figure 9.** Different methods interact with virtual objects in the third-person perspective. (**a**) Pixel avatar. (**b**) 2D video conference. (**c**) Artificial avatar. (**d**) Skeleton avatar.

Q1.  Can the avatar reflect the shape and texture of the arms of a real person?
Q2.  How was the interactive experience with the virtual objects?
Q3.  How realistic was the meeting? Was there a feeling of coexisting in the same space with other participants?

Q1 evaluates whether the avatar's arms are realistic and Q2 evaluates which avatar is the most convenient and natural for object interaction. Q3 is devoted to evaluating the performance of different avatars in terms of replicating meeting scenes in the real world. The user survey results for the three question are presented in Figure 10.

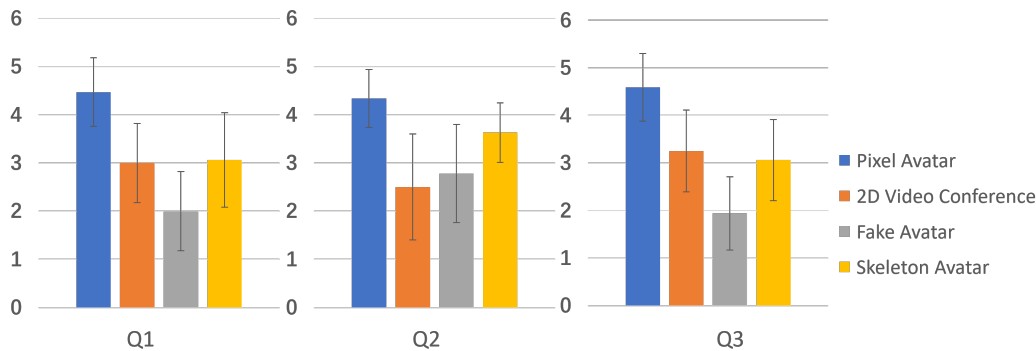

**Figure 10.** Results of the comparative experiments. The error bars in the chart represent standard deviations (SDs).

For Q1, the mean value of our method (mean = 4.47, SD = 0.71) is higher than that of the 2D video conference (mean = 3.00, SD = 0.82), artificial avatar (mean = 2.00, SD = 0.82), and skeleton avatar (mean = 3.06, SD = 0.98). Through t-test analysis, we can conclude that our method significantly outperforms the 2D video conference ($t(30) = 5.40$, $p < 0.05$), artificial avatar ($t(30) = 9.08$, $p < 0.05$), and skeleton avatar ($t(30) = 4.63$, $p < 0.05$), meaning it can better reflect user clothing and arms.

The artificial avatar uses a virtual avatar that is entirely unrelated to the user. Therefore, it cannot reflect the shape and texture of the user's arms, and its score is the lowest. Both the 2D video conference and skeleton avatar can reflect the user's arm texture because they contain information from the user's 2D video stream, but because they are not 3D, they cannot reflect the arm shape. Therefore, the difference in scores between these two methods is not significant. Our method can reflect both texture and shape characteristics, and achieved the highest score.

For Q2, which is related to the interaction experience, the results reported for our method (mean = 4.34, SD = 0.60) are significantly higher than those of the 2D video conference (mean = 2.50, SD = 1.10) ($t(30) = 5.91$, $p < 0.05$), artificial avatar (mean = 2.78, SD = 1.02) ($t(30) = 5.30$, $p < 0.05$), and skeleton avatar (mean = 3.63, SD = 0.62) ($t(30) = 3.34$, $p < 0.05$).

The 2D video conference yielded the lowest score when interacting with virtual objects because it cannot represent the interaction between the avatar's hand and the objects, so the feeling of contact is not apparent, particularly from the first-person perspective. The artificial avatar can represent the interaction between hands and virtual objects, but is limited by a lack of user image realism, so its score in the interactive experience is also lower than that of our pixel avatar. The interaction logic of the skeleton avatar is approximately the same as that of our method. The arm composed of balls and rods can accurately interact with virtual objects from both the first-person and third-person perspectives. Among the three baselines, the score of the skeleton avatar is the closest to that of our method. Compared to the skeleton avatar, our method can more accurately reflect the volume and texture of the arm from a first-person perspective, so the contact performance during interaction is more intuitive, resulting in the highest evaluation score.

For Q3, the mean score of our method is 4.59 (SD = 0.71), which is higher than those of the 2D video conference (mean = 3.25, SD = 0.86), artificial avatar (mean = 0.94, SD = 0.77), and skeleton avatar (mean = 3.06, SD = 0.85). The results of our method are also significantly higher than those of the 2D video conference ($t(30) = 4.83$, $p < 0.05$), artificial avatar ($t(30) = 10.12$, $p < 0.05$), and skeleton avatar ($t(30) = 5.51$, $p < 0.05$).

These results indicate that our method is more similar to an offline face-to-face meeting compared to the other avatar-generation methods. While observing the avatars of other users in a virtual meeting from the third-person perspective, it is difficult for users to determine the identities of other participants in a conference while using artificial avatars, which affects the feeling of coexistence when meeting other users, resulting in the lowest score. Although 2D video conferences and skeleton avatars can reflect the person who a user is meeting, 2D video conferences lack the 3D sense of real people. The skeleton avatar has a pair of repeated arms, which differ from those of real people. Our method can more accurately recreate 3D conference participants compared to the baselines while reflecting the identities of participants in a conference, resulting in the highest score for the sense of spatial coexistence.

### 5.3. Usability Experiment

Our second experiment aimed to verify the usability of the proposed method and used the same equipment and virtual meeting room scene as the first experiment. In addition to interacting with virtual objects in the same manner as in the first experiment, the second experiment also allowed participants to perform some common arm movements, such as hand waving, during online conferences and generates their pixel avatars in real-time, showing their appearance from their own perspective and other people's perspectives,

respectively. We asked users to comprehensively evaluate the performance of our method in virtual meetings and virtual object interaction and complete the system usability scale (SUS) questionnaire [44] to evaluate the usability of our method. The SUS questionnaire is widely used for interface evaluation and contains 10 items on a 5-point scale (1 to 5 points). As shown in Figure 11, the SUS score of our method is 88.28, which is much higher than the average score of 68 on the SUS questionnaire, indicating excellent feedback. From these results, we can conclude that our system is easy to understand and use, and can help users better complete tasks, such as virtual object interactions in a virtual environment.

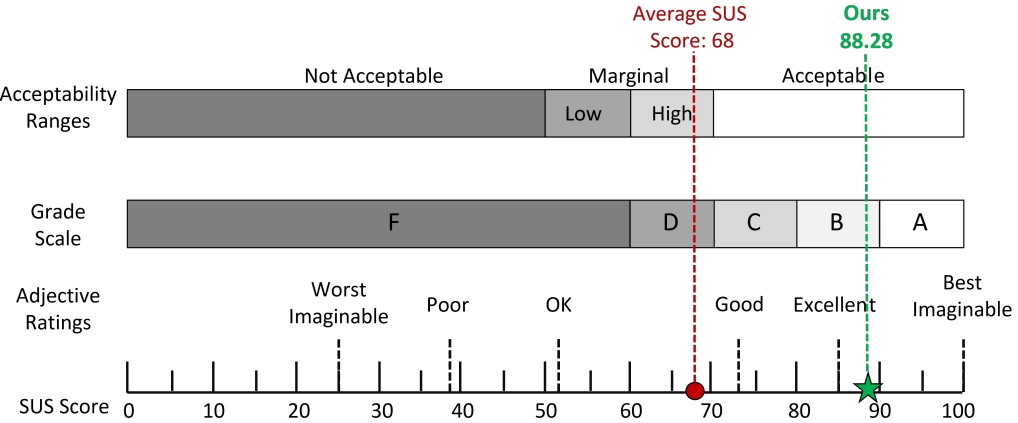

**Figure 11.** SUS score results for the proposed pixel avatar method.

Additionally, we conducted an experiment to evaluate the performance of the proposed avatar. The results demonstrate that our method can generate pixel avatars of various resolutions (see Figure 12). Additionally, we recorded the FPS of the avatars of different resolutions during a virtual meeting. The results reveal that we can generate a 320 × 240 resolution pixel avatar at 12 FPS, 160 × 120 resolution pixel avatar at 19 FPS, 80 × 60 resolution pixel avatar at 20 FPS, and 40 × 30 resolution pixel avatar at 20 FPS. While generating the images of head and body and pose data required by the pixel avatar in the human video stream processing stage, the time required for the 320 × 240 resolution pixel avatar is 40 ms, the time required for the 160 × 120 resolution pixel avatar is 38 ms, the time required for the 80 × 60 resolution is 38 ms, and the time required for 40 × 30 resolution pixel avatar is 37 ms. From left to right, the avatar resolutions range from high to low. The low-resolution results reflect the artistic pixel style, whereas the high-resolution results are closer to real people and have a sense of realism. We investigated participant preferences regarding the resolution of the generated avatars. More than 93% (15 of all 16 participants) of the participants chose the highest 320 × 240 resolution pixel avatar because it can represent more body features and details, and the higher-resolution pixel body is more consistent with the high-resolution head video stream.

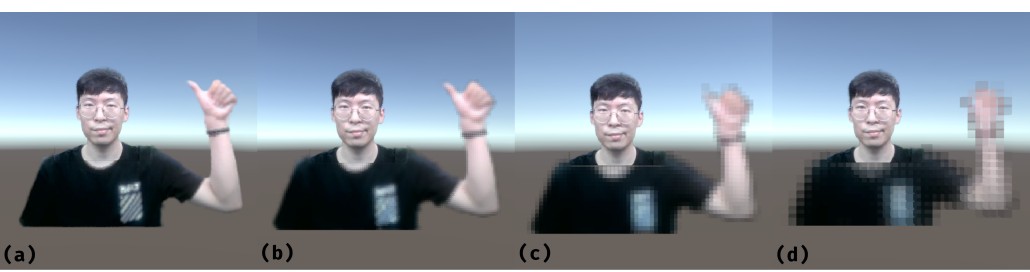

**Figure 12.** Pixel avatars with different resolutions: (**a**) 320 × 240 resolution pixel avatar; (**b**) 160 × 120 resolution pixel avatar; (**c**) 80 × 60 resolution pixel avatar; (**d**) 40 × 30 resolution pixel avatar.

## 6. Discussion

The main goal of our study is to develop an avatar that expresses the characteristics of users in 3D virtual meetings and interactions and has a good sense of contact and spatial coexistence. In a comparative experiment based on a user study considering several baseline methods, we first proved that our method has the same advantages as a 2D video conference, which can reflect the realistic features of user expression and clothing. Additionally, similar to an artificial avatar, our method has the advantage of a more direct interaction experience in a 3D virtual space. Figure 13 shows multi-person interactions with virtual objects using pixel avatars in a virtual meeting. Overall, our method is more comprehensive than the baseline methods. We also verified the simplicity and effectiveness of our system through SUS questionnaire experiments and proved the effectiveness of each component of our framework through different supplemental experiments.

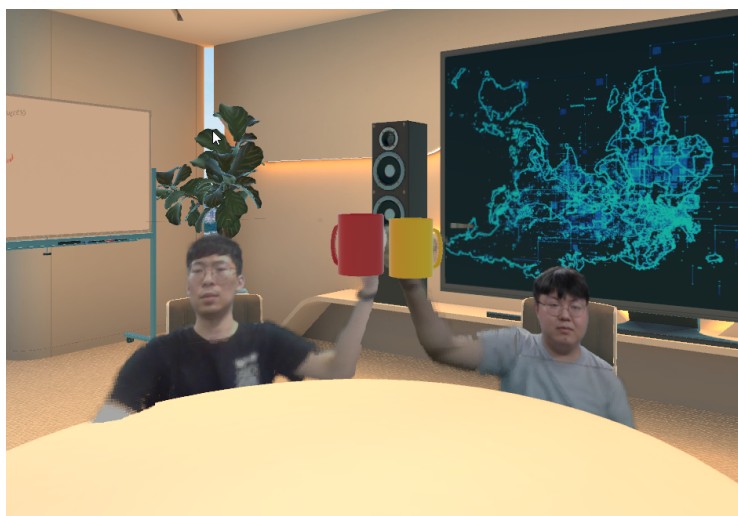

**Figure 13.** Multi-person use pixel avatars to interact with virtual objects in virtual space.

We aim to apply pixel avatars to multi-person virtual conferences and virtual interaction scenarios. Although we built a multi-person interaction scene as shown in Figure 13, we only performed single-person experiments interacting with virtual objects and real-time avatar generation experiments for consistency and objectiveness. There is no comparative experiment with other avatar types on communication efficiency through various non-verbal communication means, such as body language, in multi-person scenarios. In the future, we plan to further improve the pixel avatar and its generation method and conduct a more detailed comparison between the pixel avatar and other existing methods in multi-person scenarios.

In the avatar generation speed test, the time required to obtain the data needed to visualize the pixel avatar through the entire human video stream processing has no noticeable difference for the four avatars of different resolutions, all of which are in the range of 37 ms to 40 ms. The significant differences in the generation speed between the high-resolution avatar and the low-resolution avatar are mainly caused by the visualization part of the pixel avatar, which is the performance bottleneck of our method. While implementing avatar visualization, we represent each pixel of the body image by a particle of Unity's particle system, and the excessive particle number of $320 \times 240$ resolution avatar leads to a significant drop in speed. A more efficient rendering method to visualize pixel avatars would significantly improve the overall performance of representing avatars, which is a key part of our future work.

According to our results and analysis, the body of a low-resolution pixel avatar is closer to the style of pixel art, but it does not match a high-resolution head. In contrast, a high-resolution pixel avatar body point cloud is more consistent with a high-resolution head but loses the pixel art style of the avatar. Based on the user preferences for higher-

resolution avatars in our user survey, our current pixel-stylization method is not highly welcomed. We assumed that it is mainly caused by the resolution gap between the high-resolution head part and pixel-style body part. One possible solution is learning-based style transfer methods such as those used in [6,12]. To make the avatar head and body style more unified and vivid, we assume that the learning-based pixel-style transfer method rather than the current algorithm method can express details of both the face and body more clearly. As some users may prefer high-resolution avatars to pixel-style art, another possible solution is improving the resolution of the pixel avatar to replicate a user's skin tone and clothing more accurately. As shown in the speed test, there is no significant difference between the human video stream processing time required for high-resolution avatars and low-resolution avatars. The final performance difference is mainly caused by the visualization stage. We would be able to use our method to generate a pixel avatar of the whole body at the original high resolution to meet the needs of users who prefer high resolution if the visualization performance is improved. Therefore, improving performance is an essential part of our future work.

Additionally, the current method is limited when the hand, arm, upper arm, and lower arm occlude each other. Additionally, the arm range judgment method we adopted may lead to the misjudgment of an occluded portion of the arm. Furthermore, when a part of the arm is in front of the torso or head, our method can misjudge some parts of the body or head as the arm, leading to an erroneous arm region. Future work will aim to improve the arm partitioning approach by using a more robust method. Although this work only applied 3D pose estimation for the arm region, our method can also be easily extended and used for a full-body video stream scene, so body parts such as the legs can also be converted into three dimensions. In the future, we wish to apply 3D pose recognition to generate a full-body interactive pixel avatar.

Although there are already methods to generate 3D avatars similar to ours for the virtual environments using the depth map generated by sensors, in this work, we propose a method to generate a 3D avatar using only a single monocular RGB webcam; we use a 3D pose estimation network instead of a time-consuming depth estimation network to convert the avatar into 3D, which reduces the amount of computational complexity during generation. Unlike the mesh style used by most existing avatars, the 3D avatar we proposed is pixel style, which provides an extra choice for user in the virtual environment. In the future, the proposed pixel avatar could be extended to a full-body 3D avatar with higher resolution and various art styles, which could be widely used in 3D virtual meetings and metaverse environments.

## 7. Conclusions

In this paper, we proposed a novel pixel-style 3D virtual avatar for virtual meetings that requires only a monocular RGB camera and can be generated in real-time. Our proposed pixel avatar contains both a high-resolution face and low-resolution point cloud body representing user features, such as clothing. It meets both the needs of real-time facial expression communication through head video stream and realistic interaction in a virtual environment using 3D arms. According to [45], human visual systems take 50 ms to process a frame. Our method generated pixel avatars of 160 × 120 resolution and below at 20 FPS, which satisfy humans' processing and reaction speed for virtual interactions. The generation speed of the pixel avatar with 320 × 240 resolution is also higher than the minimum frame rate to avoid performance degradation in a virtual environment of 10 Hz proposed by [46,47]. According to experimental results, our method combines the advantages of 2D video conferences and artificial 3D avatars, compensates for the shortcomings of these methods, and obtains higher user evaluations for virtual meetings and interactions.

**Author Contributions:** Conceptualization, H.D. and Q.Z.; methodology, H.D.; software, H.D. and H.J.; validation, C.-H.K.; paper, H.D. and Q.Z. All authors have read and agreed to the published version of the manuscript.

**Funding:** This work was partly supported by Institute of Information & Communications Technology Planning & Evaluation (IITP) grant funded by the Korea government (MSIT) (No. IITP2022-0-002880101002) and National Research Foundation of Korea (NRF) grant funded by the Korea government (MSIT) (No. NRF2022R1A4A101886911).

**Institutional Review Board Statement:** Not applicable.

**Informed Consent Statement:** Not applicable.

**Data Availability Statement:** The data presented in this study are available on request from the corresponding author.

**Conflicts of Interest:** The authors declare no conflict of interest.

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
