# Peer review of "Real-Time Interaction for 3D Pixel Human in Virtual Environment"

_applsci, doi:10.3390/app13020966_

Round 1
Reviewer 1 Report
To address the challenge of achieving realistic body contact or interactive experiences, the authors present a novel pixel-style virtual avatar for interacting in virtual conferences. In contrast to a typical 3D avatar, the pixel-style avatar can provide a more efficient communication experience and express a user's natural body and clothing characteristics. The topic is very interesting and has many application scenarios.
Strengths:
1. The presented approach proposes a novel pixel-style avatar to solve segmented human real-time 3D visualization problems in a virtual environment.
2. The authors use 3D pose data to generate a 3D pixel avatar arm and use high-resolution facial video to reflect facial expressions, enabling more realistic interactions and intuitive communication.
Weakness:
1. The abstract is not well-organized; it should briefly state the research subject's general aspects and the main conclusions.
2. The literature review should be improved. The authors should summarize the advantages and disadvantages of related methods and compare their approach with the existing methods.
3. In practical scenarios of this approach, efficiency and robustness are also important. The authors should report at least the response time of the proposed approach.
4. The experimental results need to be further discussed. It's better to point out any exceptions or unsettled points.
5. I suggest discussing the theoretical implications of your work in the Discussion section.
6. In the Conclusion section, I suggest summarizing your evidence for each conclusion. For example, lines 498-499 indicate "the needs of real-time facial expression communication" how do we address evidence for "real-time" in practical scenarios?
Reviewer 2 Report
I thank the authors for the opportunity to read this article. The use of stylized avatars in VR based on the real-life appearance of participants certainly has a future. Still, I have a few reservations about this article.
* The use of these avatars is not clear. I assume the final use is an immersive VR system (currently using a VR headset). One can assume that the user's head is shielded by this device and thus we cannot capture their facial expressions, which is one motivation.
* Pixel-style art is not simply a reduction in resolution to achieve coarse pixel display. Which is very misleading, given the first sentence of the abstract. I do not consider the avatars shown to be pixel-art stylization. It would also be useful to list more methods here.
* I consider the division of the avatar into two parts with different pixel resolutions (head, rest of the body) unfortunate and the result of the testing (lines 458-460) shows this. The claim that this is an optimization is very strange and needs to be backed up by measurements.
* The method of estimating 3D hand points is rather naive (section 3.3.3). Would it not be possible to use a depth map generated for given images (e.g. DepthGAN, DGGAN, etc. ). Alternatively, at least add a comparison with similar methods.
* Considering the test PC (line 355-357) I find the speeds very low (320x240 at 12FPS). For this reason it would be useful to add a discussion of bottleneck supported by measurements to the paper.
* I positively evaluate the results of user testing and SUS skore. Still, I would have appreciated more detailed description of the user test, especially the way of non-verbal communication in an immersive environment. It is not clear from the description whether the users evaluated a representation of themselves or someone else. Whether it was only on a PC monitor or in VR.
I recommend to edit the abstract to be more in line with the content, add related works to the mentioned parts, better describe the experiments with users and describe why you don't achieve realtime-performance.
I strongly believe that my observations will contribute to the quality of the paper.
Round 2
Reviewer 2 Report
Thank you for clarifying and responding to my comments. I believe it has helped the clarity of the text.